# Cross-linking peptide and repurposed drugs inhibit both entry pathways of SARS-CoV-2

Hanjun Zhao[1,2], Kelvin K. W. To [1,2,3], Hoiyan Lam[2], Xinxin Zhou[2], Jasper Fuk-Woo Chan [1,2,3], Zheng Peng[2], Andrew C. Y. Lee [2], Jianpiao Cai[2], Wan-Mui Chan[2], Jonathan Daniel Ip[2], Chris Chung-Sing Chan[2], Man Lung Yeung[1,2,3], Anna Jinxia Zhang[1,2], Allen Wing Ho Chu[2], Shibo Jiang [4] & Kwok-Yung Yuen [1,2,3✉]

Up to date, effective antivirals have not been widely available for treating COVID-19. In this study, we identify a dual-functional cross-linking peptide 8P9R which can inhibit the two entry pathways (endocytic pathway and TMPRSS2-mediated surface pathway) of SARS-CoV-2 in cells. The endosomal acidification inhibitors (8P9R and chloroquine) can synergistically enhance the activity of arbidol, a spike-ACE2 fusion inhibitor, against SARS-CoV-2 and SARS-CoV in cells. In vivo studies indicate that 8P9R or the combination of repurposed drugs (umifenovir also known as arbidol, chloroquine and camostat which is a TMPRSS2 inhibitor), simultaneously interfering with the two entry pathways of coronaviruses, can significantly suppress SARS-CoV-2 replication in hamsters and SARS-CoV in mice. Here, we use drug combination (arbidol, chloroquine, and camostat) and a dual-functional 8P9R to demonstrate that blocking the two entry pathways of coronavirus can be a promising and achievable approach for inhibiting SARS-CoV-2 replication in vivo. Cocktail therapy of these drug combinations should be considered in treatment trials for COVID-19.

[1] State Key Laboratory of Emerging Infectious Diseases, Li Ka Shing Faculty of Medicine, The University of Hong Kong, Pokfulam, Hong Kong SAR, China. [2] Department of Microbiology, Li Ka Shing Faculty of Medicine, The University of Hong Kong, Pokfulam, Hong Kong SAR, China. [3] Carol Yu Centre for Infection, Li Ka Shing Faculty of Medicine, The University of Hong Kong, Pokfulam, Hong Kong SAR, China. [4] Key Laboratory of Medical Molecular Virology (MOE/NHC/CAMS), School of Basic Medical Sciences, Fudan University, Shanghai, China. ✉email: kyyuen@hku.hk

The COVID-19 pandemic is a devastating global health threat of this century. There is not yet a reliable antiviral available for therapy or prevention of SARS-CoV-2 infection. Studies showed that the COVID-19 patients may have decreasing level of antibodies[1–5], which suggested that SARS-CoV-2 vaccine may also have varying duration of protection among different individuals. Furthermore, reports of re-infection hinted that the immune responses to SARS-CoV-2 might not sufficiently protect some patients from re-infection by SARS-CoV-2[6]. The antibody-dependent enhancement is another potential side effect of SARS-CoV-2 vaccines[7,8]. Broad-spectrum antivirals, not relying on host immune responses against viruses, are urgently needed for treating COVID-19 and other coronavirus infections. Thus, broad spectrum antiviral peptides against SARS-CoV-2[9,10] and repurposing of FDA-approved drugs are studied for the inhibition of SARS-CoV-2[11–13].

Since the emergence of COVID-19, many clinical trials have been carried out for repurposing the approved drugs including chloroquine, arbidol, camostat, remdesivir, ribavirin, and lopinavir/ritonavir against SARS-CoV-2[14]. Chloroquine probably interfered with endocytic pathway to broadly inhibit SARS-CoV-2[15], SARS-CoV[16], influenza virus, Ebola and other viruses in vitro[17]. However, its clinical efficacy is limited in COVID-19 patients[18–20] and its potential cardiac side effects and lack of antiviral activity in vivo are major concerns[12,21]. Umifenovir also known as arbidol, the clinically available drug in China and Russia, is in Phase III trial against influenza in US. Arbidol demonstrated broad-spectrum in vitro antiviral activity against many viruses including influenza virus, coronaviruses, and Ebola[22,23], with an $IC_{50}$ of 2-20 µg ml$^{-1}$ against SARS-CoV-2[15,24]. However, the peak serum concentration of arbidol is lower than 2 µg ml$^{-1}$ within 5 h after administration of usual drug dosage[25,26], which might explain the uncertain clinical efficacy of arbidol in SARS-CoV-2 patients[27–29]. Camostat mesylate (Camostat), the inhibitor of TMPRSS2 which facilitates virus entry on cell surface, has been showed to inhibit SARS-CoV, SARS-CoV-2 and other viruses[30,31]. Since ACE2 and TMPRSS2 are individually expressed in some human cell types or co-expressed in other cell types[32], the approach of simultaneous inhibition of virus entry through the endocytic pathway and the surface fusion pathway mediated by TMPRSS2 may have better antiviral effect.

In this study, a cross-linking peptide 8P9R, which is developed from our previously reported P9[33] and P9R[10], has been shown to have dual-antiviral mechanisms of cross-linking viruses to stop viral entry (mediated by TMPRSS2 for SARS-CoV-2) and of reducing endosomal acidification to inhibit viral entry through endocytic pathway. 8P9R shows significantly antiviral activity against SARS-CoV-2 in hamsters and SARS-CoV in mice. Moreover, we try to identify clinical drug combinations which can inhibit two entry pathways of SARS-CoV-2 to efficiently inhibit viral replication in vivo. We demonstrate that endosomal acidification inhibitors (8P9R or chloroquine) can significantly enhance the antiviral efficiency of arbidol, which is found to inhibit virus-cell membrane fusion, at a clinically achievable concentration against SARS-CoV-2 and SARS-CoV replication in Vero-E6 cells, where coronaviruses mainly enter cells through endocytic pathway. The synergistic mechanism study indicates that 8P9R or chloroquine can elevate endosomal pH which enhances the efficiency of arbidol in blocking virus-host cell fusion mediated by spike and ACE2. To block the two entry pathways of coronavirus, arbidol and chloroquine were combined with camostat which inhibits TMPRSS2 to prevent SARS-CoV-2 fusion on cell surface. Results show significant antiviral activity against SARS-CoV-2 in hamsters and SARS-CoV in mice. This drug combination has a similar inhibitory effect as the dual-functional 8P9R in the treatment of SARS-CoV-2 and SARS-CoV animal models. In contrast, the single use of arbidol or chloroquine does not show any antiviral efficacy in mice and hamsters. Given that all these three drugs are broad-spectrum antivirals, this combination may play important roles in controlling respiratory virus infection with similar entry pathways. The identification of the dual-functional 8P9R and the triple combination of clinical drugs prove that targeting both entry pathways of coronavirus can be a feasible approach to inhibit SARS-CoV-2 replication in vivo.

## Results

**8P9R showed potent antiviral activity against SARS-CoV-2.** We previously showed that a broad-spectrum antiviral peptide P9R could suppress coronavirus and influenza virus by binding to viruses and inhibiting virus-host endosomal acidification[10]. We hypothesized that if single unbranced P9R could bind to virus surface and capture viruses, then the branched P9R could cross-link viruses (Fig. 1a) to enhance the antiviral activity. First, we measured the binding ability of eight-branched P9R (8P9R) and single P9R to SARS-CoV-2 and H1N1 virus by measuring the RNA copies of viruses binding to ELISA plate, on which peptides were coated. The viral RNA copies indicated that 8P9R could efficiently bind to viruses and capture viral particles on ELISA plate when compared with BSA and P9RS[10] which was a basic peptide with no antiviral activity or binding ability to virus (Fig. 1b). This 8P9R suppressed SARS-CoV-2 infection more potently than P9R when viruses were pretreated by peptides (Fig. 1c), treated during viral inoculation (Fig. 1d) or post-infection (Fig. 1e). 8P9R showed more potent antiviral activity ($IC_{50} = 0.3$ µg ml$^{-1}$) in high salt condition (PBS) than that ($IC_{50} = 20.2$ µg ml$^{-1}$) of P9R in PBS (Fig. 1b), even though P9R showed potent antiviral activity ($IC_{50} = 0.9$ µg ml$^{-1}$) in low salt concentration of 30 mM phosphate buffer (PB) (Supplementary Fig. 1). This is consistent with a previous report that antimicrobial activities of defensins are sensitive to high salt condition[34]. Furthermore, no obvious hemolysis was observed when turkey red blood cells were treated by 8P9R at 200 µg ml$^{-1}$ (Fig. 1f) and the cytotoxicity assay indicated that $TC_{50}$ of 8P9R was higher than 200 µg ml$^{-1}$ in Vero-E6 cells (Supplementary Fig. 2).

**The dual-functional activities of 8P9R against virus.** To demonstrate the cross-linking ability, TEM images were taken to show that 8P9R could cross-link SARS-CoV-2 to form big viral cluster (Fig. 2a and Supplementary Fig. 3). In contrast, the peptide P9RS without binding ability (Fig. 1b) and single P9R did not cross-link virus to form big viral cluster. We further confirmed this result with fluorescence-labelled H1N1 virus (Fig. 2b and Supplementary Fig. 4). The confocal pictures showed that 8P9R could efficiently cross-link H1N1 viruses that were aggregated around the cell membrane without entry when compared with the treatment of P9RS or P9R. Furthermore, we demonstrated that 8P9R could efficiently inhibit endosomal acidification (Fig. 2c), which was similar to the endosomal acidification inhibitor bafilomycin A1. These results indicated the dual-functional activities of 8P9R, which inhibited endosomal acidification required in endocytic pathway of viral infection and cross-linked viruses on the cell membrane surface without entry. The cross-linked viruses might affect SARS-CoV-2 entry on cell surface through TMPRSS2-mediated pathway. Thus, we further confirmed that 8P9R could inhibit SARS-CoV-2 infection through TMPRSS2-mediated surface entry pathway in Calu-3 cells in the later section.

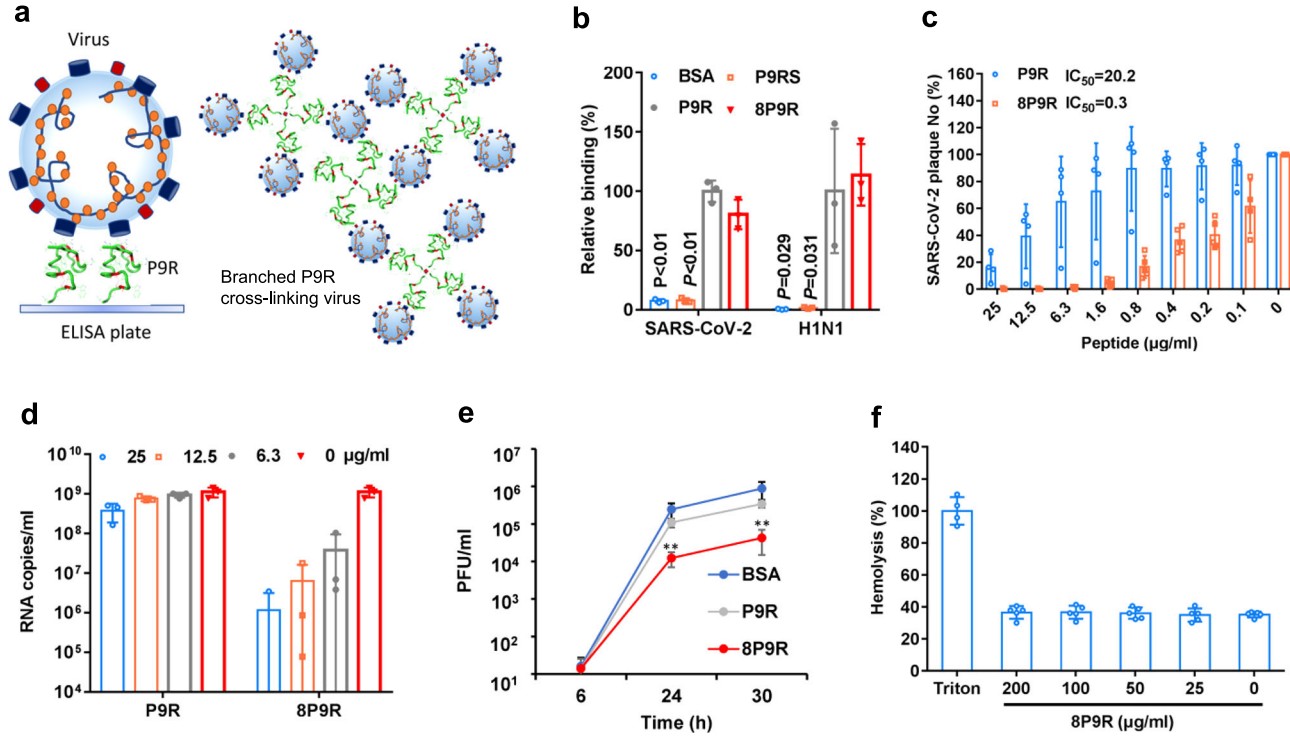

**Fig. 1 The enhanced antiviral activity of branched P9R (8P9R). a** The schematic figure of single P9R binding to single viral particle and branched P9R (8P9R) cross-linking viruses together. **b** The binding of 8P9R and P9R to SARS-CoV-2 and H1N1 viruses. Peptides coated on ELISA plates could capture virus particles which were then quantified by RT-qPCR. P9RS was the negative control peptide with no viral binding ability. Relative binding and $P$ values were compared to P9R. Data are presented as mean ± SD of three independent experiments. **c** SARS-CoV-2 was pretreated with the indicated peptides for plaque reduction assay. Data are presented as mean ± SD of four independent experiments. **d** SARS-CoV-2 was treated by indicated peptides (25 μg ml$^{-1}$) during viral inoculation. Viral RNA copies were detected by RT-qPCR at 24 h post infection in the supernatant of Vero-E6 cells. Data are presented as mean ± SD of three independent experiments. **e** SARS-CoV-2 was treated by peptides (50 μg ml$^{-1}$) at 6 h post infection. Viral titers were measured at the indicated time by plaque assay. Data are presented as mean ± SD of three independent experiments. $P$ values were compared with BSA. **f** Hemolysis assay of 8P9R in turkey red blood cells (TRBC). TRBC were treated by the indicated concentration of 8P9R. Hemolysis (%) was normalized to TRBC treated by Triton X-100. Data are presented as mean ± SD three independent experiments. $P$ values are calculated by two-tailed student $t$ test. ** indicates $P < 0.01$. Source data are provided as a Source Data file.

**8P9R could enhance arbidol at low concentration to inhibit SARS-CoV-2.** Serial monitoring by viral load and sequencing of clinical samples from COVID-19 patients showed that SARS-CoV-2 could be detected for more than 1 month with occasional detection of mutants[35,36]. These findings suggested potentially low sterilizing efficiency of human immune response for clearing SARS-CoV-2 in some patients. Thus, the repurposing of the anti-influenza drug arbidol available in China and Russia was considered. Arbidol showed in vitro antiviral activity against coronaviruses including SARS-CoV-2 and SARS-CoV. However, its relatively low serum concentration in human bodies[25,26] may account for its poor antiviral efficacy in patients[28,29]. We showed that 8P9R significantly enhances the antiviral efficiency of arbidol at the concentration lower than the normal IC$_{50}$ (3.6 μg ml$^{-1}$) of arbidol (Fig. 3a). Importantly, 8P9R could elevate the antiviral activity of arbidol at low concentration (0.2 μg ml$^{-1}$) when arbidol itself did not show antiviral activity (Fig. 3b and Supplementary Fig. 5). This low concentration is closer or even lower than the concentration of arbidol in human serum.

**The synergistic mechanism of 8P9R enhancing arbidol against SARS-CoV-2.** To determine the synergistic enhancing mechanism of 8P9R on arbidol to inhibit SARS-CoV-2, we firstly clarified that arbidol could slightly reduce viral attachment (Supplementary Fig. 6). Next, when viruses (10$^6$ PFU ml$^{-1}$) was pretreated by arbidol (25 μg ml$^{-1}$) and then diluted to 10,000 folds for plaque assay, arbidol did not inhibit SARS-CoV-2 infection (Fig. 3c). In

contrast, 8P9R could significantly reduce the number of infectious viruses even with >1000-fold dilution, which indicated that the antiviral activity of 8P9R depended on targeting virus (Fig. 3c), similar to P9R[10]. We further showed that arbidol could significantly inhibit SARS-CoV-2 replication after viral entry in the time of addition experiment as that by bafilomycin A1, a known host targeting antiviral to inhibit cell endosomal acidification. (Fig. 3d). These results indicated that the main target of arbidol against SARS-CoV-2 is host cells, but not the virus. Next, we demonstrated that arbidol could efficiently inhibit spike-ACE2 mediated cell-cell fusion in 293T cells (Fig. 3e) and Huh-7 cells (Supplementary Fig. 7), which indicated that arbidol could inhibit virus-cell membrane fusion. The fusion inhibition of arbidol on SARS-CoV-2 was consistent with the claim that arbidol could block the release of SARS-CoV-2 in endolysosomes[24]. Since lysosomes are the fusion location of SARS-CoV-2 infection through endocytic pathway[37] and the endosomal acidification inhibitors, ammonium chloride[30], bafilomycin A1 and 8P9R (125 μg ml$^{-1}$) could inhibit spike-ACE2 mediated cell membrane fusion (Fig. 3e), we suspected that the pH in endosomes/lysosomes could affect the inhibition efficiency of arbidol on spike-ACE2 mediated fusion. Using a low concentration of 8P9R combined with the low concentration of arbidol could more efficiently block the spike-ACE2-mediated membrane fusion (Fig. 3e) when compared with 8P9R or arbidol alone at 25 μg ml$^{-1}$. Thus, the mechanism of synergistic enhancement of arbidol by 8P9R is due to the inhibition of endosomal acidification by 8P9R, so that

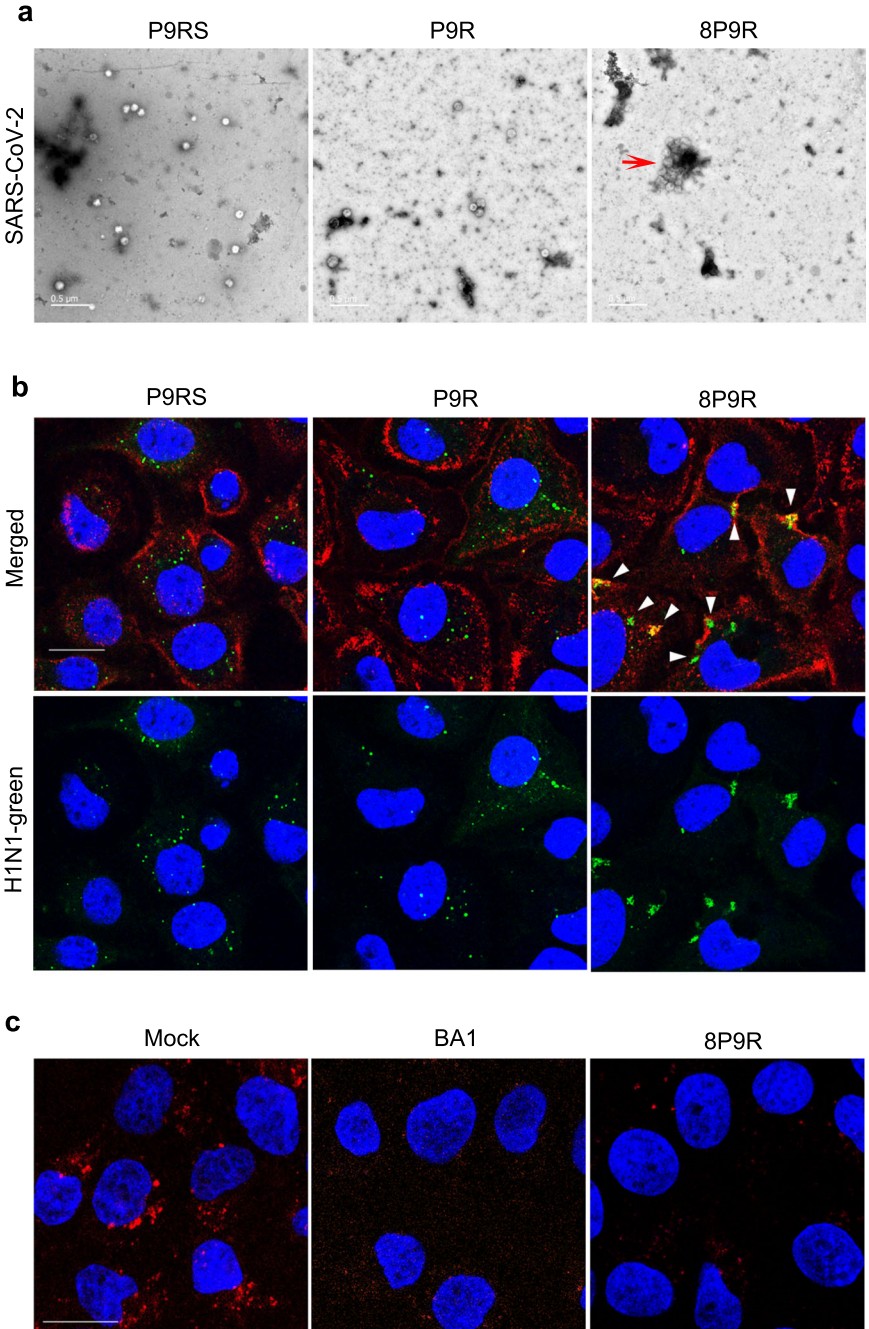

**Fig. 2 The dual-functional activities of 8P9R. a** Cross-linking of SARS-CoV-2 by 8P9R. SARS-CoV-2 was treated by 8P9R, P9R, or P9RS (50 μg ml⁻¹). The treated virus was negatively stained for TEM assay. The red triangle indicates the big cluster of cross-linked SARS-CoV-2. Scale bar = 0.5 μm. For quantification, 55 independent viral particles of P9RS-treated virus, 50 independent viral particles of P9R-treated virus, and 13 viral particles (including independent and clustered particles) of 8P9R-treated virus could be accounted in 5 representative microscope fields. The big clustering viral particles in 8P9R-treated samples could be more than 500 nm, which was bigger than the size (~100 nm) of the usual SARS-CoV-2 virion. **b** H1N1 virus was pre-labelled by green fluorescence dye and then treated by peptides. After 1 h infection in MDCK cells, cells were fixed and stained by cell membrane dye (red) and nuclear dye (blue). White triangles indicate the cross-linked viruses located at cell membrane. Scale bar = 20 μm (**c**). 8P9R could efficiently inhibit endosomal acidification. MDCK cells were treated by 8P9R (25 μg ml⁻¹), bafilomycin A1 (BA1, 50 nM), BSA (Mock) and low pH indicator pHrodo™ Red dextran. Red dots indicate the endosomes with low pH. Nuclei were stained with nuclear dye (blue). Live cell images were taken by confocal microscopes. Scale bar = 20 μm. Experiments were repeated twice independently. Source data are provided as a Source Data file.

arbidol could more efficiently inhibit virus-cell fusion at the higher pH environment.

**Endosomal acidification inhibitors enhance arbidol against coronaviruses**. To further confirm the endosomal acidification inhibitors can synergistically enhance the antiviral activity of

arbidol and to find clinically available drug for inhibiting SARS-CoV-2, we identified that chloroquine, a known drug elevating endosomal pH, could significantly enhance the antiviral activity of arbidol at low concentrations (0.2–0.4 μg ml⁻¹) against SARS-CoV-2 (Fig. 4a) and SARS-CoV in Vero-E6 cells (Fig. 4b). Chloroquine supplemented with the low concentration of arbidol

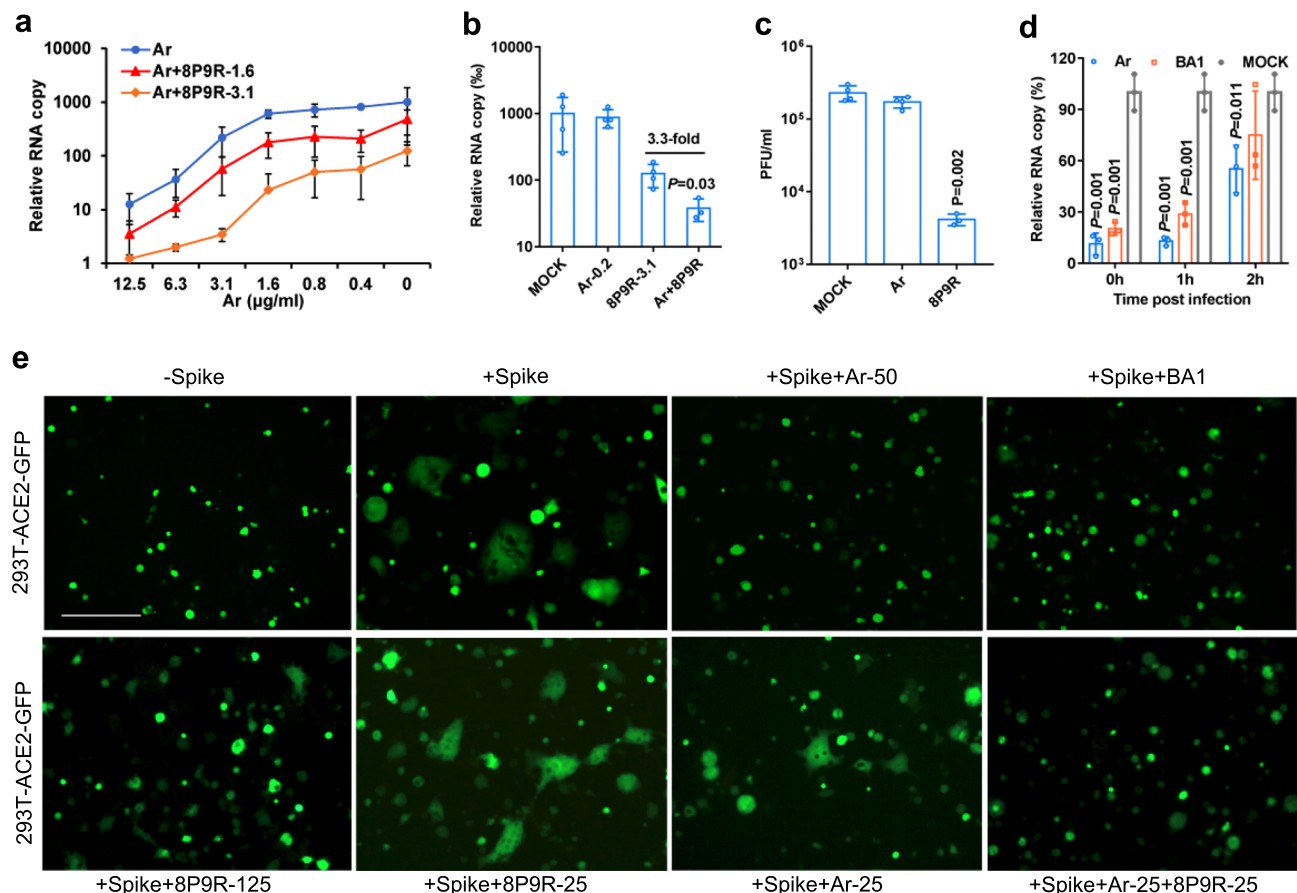

**Fig. 3 Synergistic mechanism of 8P9R enhancing the antiviral activity of arbidol. a** 8P9R could enhance the antiviral activity of arbidol against SARS-CoV-2 in Vero-E6 cells ($n = 5$). Virus infected cells at the presence of the indicated concentrations of arbidol (Ar) or Ar+8P9R (1.6 µg ml$^{-1}$) or Ar+8P9R (3.1 µg ml$^{-1}$). **b** 8P9R could significantly enhance the antiviral activity of arbidol when arbidol alone did not show antiviral activity ($n = 4$). SARS-CoV-2 was treated by the indicated Ar-0.2 (0.2 µg m l$^{-1}$), 8P9R-3.1 (3.1 µg ml$^{-1}$), Ar+8P9R, or PBS (Mock). $P$ value was compared with Ar+8P9R. **c** SARS-CoV-2 ($10^6$ PFU ml$^{-1}$) were treated by 25 µg ml$^{-1}$ arbidol, or 8P9R ($n = 3$). Then virus was serially diluted to detect the viral titer by plaque assay. **d** SARS-CoV-2 was treated at the indicated time of post infection by the indicated drugs ($n = 3$). Viral titers (**a**, **b** and **d**) were measured by RT-qPCR at 24 h post infection. Relative RNA copy was normalized to MOCK. Data in (**a**–**d**) are presented as mean ± SD from 3–5 independent experiments. $P$ values in (**b**–**d**) were calculated by two-tailed student $t$ test when compared with mock. **e** Spike-ACE2 mediated cell-cell fusion could be blocked by arbidol and endosomal acidification inhibitors (bafilomycin A1 and 8P9R). The 293T cells expressed ACE2 or spike+GFP were co-cultured at the presence of indicated 8P9R (125 or 25 µg/ml), arbidol (50 or 25 µg ml$^{-1}$) or bafilomycin A1 (BA1, 50 nM). 8P9R (25 µg ml$^{-1}$) and arbidol (25 µg ml$^{-1}$) did not block cell fusion, of which the fused cells were (2–10)-fold bigger than the non-fused cells. The 293T-GFP cells without spike (-Spike) served as the negative control of cell-cell fusion. Scale bar = 100 µm. The representative pictures were taken at 8 h after co-culture. Experiments were repeated three times independently. Source data are provided as a Source Data file.

could inhibit more than two-fold viral replication when compared with chloroquine alone (Fig. 4a–b). The combination of chloroquine and arbidol could more effectively inhibit spike-ACE2 mediated cell-cell membrane fusion (Supplementary Fig. 8), which further confirmed that endosomal acidification inhibitors elevating pH in endosomes/lysosomes could enhance the antiviral activity of arbidol by blocking virus-cell membrane fusion. Our findings support the combination of arbidol with chloroquine for better antiviral activity.

**Simultaneous blockage of the two entry pathways of coronavirus for antiviral treatment in vivo.** To test the antiviral efficacy in vivo, we challenged 10-month-old mice with SARS-CoV and then drugs were initially administrated to mice at 8 h post infection. The drug toxicity assay in mice indicated that 8P9R and combination of three drugs did not induce obvious toxicity in mouse lungs as indicated by H&E staining and body-weight monitoring for 18 days (Supplementary Fig. 9a, b).

Arbidol (30 mg kg$^{-1}$), chloroquine (40 mg kg$^{-1}$) or the combination of arbidol with chloroquine could not inhibit SARS-CoV replication in mouse lungs (Fig. 4c). The dual-functional peptide 8P9R could significantly inhibit SARS-CoV replication in mouse lungs (Fig. 4c). As expected (Fig. 4d), we showed that arbidol and chloroquine could significantly inhibit SARS-CoV-2 replication in Vero-E6 cells (without TMPRSS2[38]), but not in Calu-3 cells in which SARS-CoV-2 enters cells depending on TMPRSS2-mediated pathway[39] (Fig. 4e). However, 8P9R could significantly inhibit SARS-CoV-2 in both Vero-E6 and Calu-3 cells (Fig. 4d, e and Supplementary Fig. 10), which suggested that 8P9R not only inhibited the viral infection through endocytic pathway in Vero-E6 cells but also inhibited viral entry through TMPRSS2-mediated pathway in Calu-3 cells. We further demonstrated that camostat but not 8P9R could inhibit TMPRSS2 activity to cleave S protein (Supplementary Fig. 11), which indicated that the inhibition of SARS-CoV-2 replication in Calu-3 cells was not related to the inhibition of TMPRSS2 activity. The inhibition of SARS-CoV-2 replication in Calu-3 cells

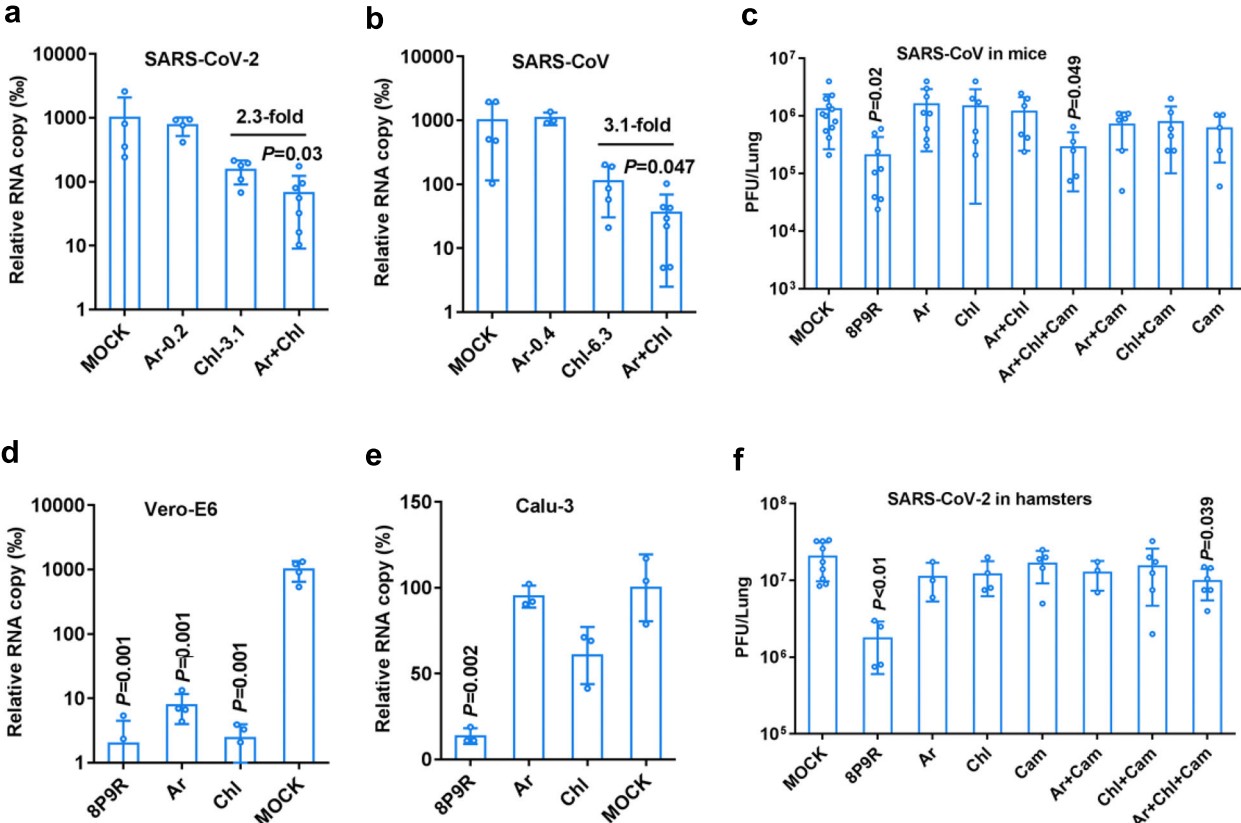

**Fig. 4 Drug combination enhanced the antiviral activity against SARS-CoV-2 and SARS-CoV. a** Chloroquine (Chl) could significantly enhance the activity of arbidol against SARS-CoV-2 while arbidol alone (0.2 µg ml$^{-1}$, Ar-0.2) did not show antiviral activity (n = 4). SARS-CoV-2 was treated by the indicated Ar-0.2, Chl-3.1 (3.1 µg ml$^{-1}$), or Ar+Chl. P value was compared with Chl-3.1. **b** Chloroquine (Chl) could significantly enhance the activity of arbidol against SARS-CoV while arbidol alone (0.4 µg ml$^{-1}$, Ar-0.4) did not show antiviral activity (n = 4). SARS-CoV was treated by the indicated Ar-0.4, Chl-6.3 (6.3 µg ml$^{-1}$), or Ar+Chl. Viral RNA copies were measured at 24 h post infection in cell supernatants. The relative RNA copy was compared to mock treated virus. P value was compared with Chl-6.3. **c** The antiviral activity of indicated drugs or drug combinations against SARS-CoV in mice. Mice were intranasally inoculated with SARS-CoV (5 × 10$^3$ PFU). 8P9R (intranasal 0.5 mg kg$^{-1}$, n = 8), arbidol (Ar, oral 30 mg kg$^{-1}$, n = 8), chloroquine (Chl, oral 40 mg kg$^{-1}$, n = 6), camostat (Cam, intranasal 0.3 mg kg$^{-1}$, n = 5), Ar+Chl (n = 6), Ar+Cam (n = 6), Chl+Cam (n = 6), Ar+Chl+Cam (n = 5) and mock (n = 12) were given to mice at 8 h post infection. Two more doses were given to mice in the following day. Viral loads in lung tissues were measured by plaque assay at day 2 post infection. **d–e** The antiviral activity of 8P9R (12.5 µg ml$^{-1}$), arbidol (12.5 µg ml$^{-1}$), and chloroquine (12.5 µg ml$^{-1}$) in Vero-E6 (**d**, n = 4) and Calu-3 (**e**, n = 3) cells. Viral RNA copies in cell supernatants were measured by RT-qPCR at 24 h post infection. Relative RNA copy was normalized to mock. **f** The antiviral activity of indicated drugs or drug combinations against SARS-CoV-2 in hamsters. Hamsters were intranasally inoculated with SARS-CoV-2 (5 × 10$^3$ PFU). Mock (n = 9), 8P9R (intranasal 0.5 mg kg$^{-1}$, n = 4), Ar+Chl+Cam (n = 6), Chl+Cam (n = 6), Ar+Cam (3), Cam (intranasal 0.3 mg kg$^{-1}$, n = 5), Ar (oral 30 mg kg$^{-1}$, n = 3), and Chl (oral 40 mg kg$^{-1}$, n = 4) were given to hamsters at 8 h post infection. Two more doses were given to hamsters in the following day. Viral loads in lung tissues were measured by plaque assay at day 2 post infection. Data are presented as mean ± SD of independent biological samples. P values are calculated by two-tailed student t test when compared with mock. Source data are provided as a Source Data file.

by 8P9R was attributed to the physical clustering of viral particles, which resulted in blocking of viral entry (Supplementary Fig. 12). The potent antiviral activity of 8P9R in Vero-E6, Calu-3 cells and in mouse model indicated that the simultaneous blockage of both entry pathways might more efficiently inhibit coronavirus replication in vivo. Camostat, a TMPRSS2 inhibitor, could significantly inhibit SARS-CoV-2 replication in Calu-3 cells[30], but could not inhibit SARS-CoV-2 replication and pseudotyped-particle entry in Vero-E6 cells[30,39]. Thus, we treated SARS-CoV-infected mice with the triple drug combination of arbidol, chloroquine and camostat. This combination showed potent antiviral activity against SARS-CoV in mice (Fig. 4c), similar to the antiviral activity of 8P9R, whereas the double drug combinations (arbidol+camostat or chloroquine+camostat) or camostat alone could not inhibit viral replication when compared with mock (Fig. 4c and Supplementary Fig. 13). In parallel, we further confirmed this in vivo result by treating SARS-CoV-2-infected

hamsters with different drug combinations. Viral loads in hamster lungs showed that 8P9R or the triple combination of arbidol, chloroquine and camostat could significantly inhibit SARS-CoV-2 replication when compared with mock (Fig. 4f and Supplementary Fig. 14). Arbidol, chloroquine, or camostat alone, and camostat combined with chloroquine (Fig. 4f) could not significantly inhibit SARS-CoV-2 replication in hamsters. These findings confirmed the limited clinical efficacy of arbidol or chloroquine alone for treating SARS-CoV-2 in patients. More importantly, these results provided the evidences of using endosomal acidification inhibitors (8P9R or chloroquine) to enhance the antiviral activity of arbidol against SARS-CoV-2 infection through endocytic pathway. Moreover, dual-functional 8P9R or the triple drug combination of arbidol, chloroquine and camostat can effectively block the two entry pathways of coronavirus, which translates into significant reduction of viral replication in vivo.

## Discussion

In this study, we developed a dual-functional antiviral peptide 8P9R which could cross-link viruses to block viral entry on cell surface through the TMPRSS2-mediated pathway and simultaneously inhibited endosomal acidification to block viral entry through endocytic pathway. We demonstrated the synergistic antiviral mechanism of endosomal acidification inhibitors (8P9R and chloroquine) on enhancing the activity of arbidol against SARS-CoV-2 and SARS-CoV infection through the endocytic pathway. Moreover, we provided the evidences of using the triple combination of arbidol, chloroquine and camostat, which are currently available clinical drugs, for the suppression of SARS-CoV-2 replication in hamsters and SARS-CoV in mice. Both the triple drug combination and 8P9R could significantly inhibit SARS-CoV-2 and SARS-CoV in vivo, which suggested that blocking the two entry pathways of coronavirus infection is a promising approach for treating COVID-19.

SARS-CoV-2 and SARS-CoV can infect host cells by either TMPRSS2-mediated pathway or endocytic pathway. Recent studies indicated that chloroquine did not inhibit SARS-CoV-2 replication in Calu-3 cells[39] and camostat did not inhibit SARS-CoV-2 replication in Vero-E6 cells[30]. By using a multi-targeting drug or drug combination to block the both entry pathways of coronavirus infection might be more efficient in inhibiting viral replication in patients because different human cells could express ACE2 and TMPRSS2 separately or simultaneously[32]. We demonstrated that endosomal acidification inhibitors (chloroquine and 8P9R) could synergistically enhance the antiviral activity of arbidol against SARS-CoV-2 and SARS-CoV. The synergistic mechanism was inferred that endosomal acidification inhibitors, by elevating endosomal pH, could enhance the activity of arbidol in blocking the spike-ACE2-mediated membrane fusion (Fig. 3e and Supplementary Fig. 9), which was consistent with the finding that spike-ACE2-mediated pseudotyped-particle entry was significantly affected by pH (ammonium chloride) in 293T cells[30]. However, the combination of chloroquine with arbidol did not show antiviral activity against SARS-CoV-2 and SARS-CoV in hamsters and mice. The one possible reason is that chloroquine and arbidol can only inhibit SARS-CoV-2 replication by interfering with the endocytic pathway, but not the TMPRSS2-mediated pathway (Fig. 4d–e). In contrast, 8P9R could significantly inhibit coronaviruses in vivo. 8P9R not only blocked the endocytic pathway by preventing endosomal acidification, but also caused the formation of big clumps of aggregated viral particles which could no longer enter cells by direct membrane fusion in Calu-3 cells (Supplementary Fig. 12). The combination of chloroquine and camostat could not significantly inhibit both viruses in vivo, which is probably due to the marginal antiviral activity of chloroquine on inhibiting viral infection through endocytic pathway in mice, hamsters and ferrets[21,40]. The combination of arbidol with chloroquine could more efficiently inhibit viral infection through endocytic pathway in TMPRSS2-deficient Vero-E6 cells (Fig. 4a, b). Thus, the triple combination of arbidol, chloroquine and camostat could significantly inhibit both SARS-CoV-2 and SARS-CoV replication in hamsters and mice (Fig. 4d, g) through simultaneous blockage of both entry pathways. Furthermore, these drugs are harnessing the host factors to interfere with viral replication which may therefore be less prone to induce drug resistant viral mutants.

With the widespread circulation of SARS-CoV-2 during the COVID-19 pandemic, the emergence of virus mutants and the decreasing antibody titers after recovery should alert us to the possibility of re-infection. The development of broad-spectrum antivirals is urgently needed for SARS-CoV-2 and new emerging viruses. Here, we identified the antiviral peptide 8P9R with dual functions to inhibit viral infection by cross-linking viruses to reduce viral entry on cell surface (i.e., TMPRSS2-mediated entry pathway for SARS-CoV) and by interfering endosomal acidification to block viral entry through endocytic pathway. Furthermore, our data supported the use of combination drug treatment with currently available broad-spectrum drugs (arbidol, chloroquine and camostat) to block both entry pathways of SARS-CoV-2, which could be also the potential therapeutics for other respiratory viruses. Further clinical trials with this cocktail therapy to evaluate their antiviral efficiency in COVID-19 patients and other viral infectious diseases are warranted.

## Methods

**Cells and viruses**. Madin Darby canine kidney (MDCK, CCL-34), Vero-E6 (CRL-1586), Calu-3 (HTB-55) and 293T (CRL-3216) cells obtained from ATCC (Manassas, VA, USA) were cultured in Dulbecco minimal essential medium (DMEM for Vero-E6 cells and 293T), MEM (for MDCK cells) or DMEM-F12 (for Calu-3 cells) supplemented with 10% fetal bovine serum (FBS), 100 IU ml$^{-1}$ penicillin and 100 µg ml$^{-1}$ streptomycin. The virus strains used in this study included 2019 new coronavirus (SARS-CoV-2) with a deletion in S between 23598-23627 bp when compared with the original HKU001a[41], SARS-CoV[33], and A/Hong Kong/415742/2009[42].

**Plaque reduction assay**. Peptides (P9R, P9RS and 8P9R) were synthesized by ChinaPeptide. Antiviral activity of peptides was measured using a plaque reduction assay. Briefly, peptides were dissolved in PBS or 30 mM PB containing 24.6 mM Na$_2$HPO$_4$ and 5.6 mM KH$_2$PO$_4$ at a pH of 7.4. Peptides or bovine serum albumin (BSA, 0.2–25.0 µg ml$^{-1}$) were premixed with 50 PFU of coronavirus (SARS-CoV-2) in PBS or PB at room temperature. After 45–60 min of incubation, peptide-virus mixture was transferred to Vero-E6 cells, correspondingly. At 1 h post infection, infectious media were removed and 1% low melting agar was added to cells. Cells were fixed using 4% formalin at 3 day post infection. Crystal blue (0.1%) was added for staining, and the number of plaques was counted.

**Antiviral multicycle growth assay**. SARS-CoV-2 and SARS-CoV infected Vero-E6 (0.005 MOI) or Calu-3 (0.05 MOI) cells at the presence of drugs or with the supplemental drugs at indicated post infection time. After 1 h infection, infectious media were removed and fresh media with supplemental drugs were added to infected cells for virus culture. At 24 h post infection, the supernatants of infected cells were collected for plaque assay or RT-qPCR assay.

**Viral RNA extraction and RT-qPCR**. Viral RNA was extracted by Viral RNA Mini Kit (QIAGEN, Cat# 52906, USA) according to the manufacturer's instructions. Extracted RNA was reverse transcribed to cDNA using PrimeScript II 1st Strand cDNA synthesis Kit (Takara, Cat# 6210 A) using GeneAmp® PCR system 9700 (Applied Biosystems, USA). The cDNA was then amplified using specific primers (Supplementary Table 1) for detecting SARS-CoV-2 and SARS-CoV using Light-Cycle® 480 SYBR Green I Master (Roach, USA). For quantitation, tenfold serial dilutions of standard plasmid equivalent to 10$^1$–10$^6$ copies per reaction were prepared to generate the calibration curve. Real-time qPCR experiments were performed using LightCycler® 96 system (Roche, USA).

**Hemolysis assay**. Two-fold diluted peptides in PBS were incubated with turkey red blood cells for 1 h at 37 °C. PBS was used as a 0% lysis control and 0.1% Triton X-100 as 100% lysis control. Plates were centrifuged at 350 g for 3 min to pellet non-lysed red blood cells. Supernatants used to measure hemoglobin release were detected by absorbance at 450 nm[10].

**Cytotoxicity assay**. Cytotoxicity of peptides was determined by the detection of 50% cytotoxic concentration (CC$_{50}$) using a tetrazolium-based colorimetric MTT assay[43]. Vero-E6 cells were seeded in 96-well cell culture plate at an initial density of 2 × 10$^4$ cells per well in DMEM supplemented with 10% FBS and incubated for overnight. Cell culture media were removed and then DMEM supplemented with various concentrations of peptides and 1% FBS were added to each well. After 24 h incubation at 37 °C, MTT solution (5 mg ml$^{-1}$, 10 µl per well) was added to each well for incubation at 37 °C for 4 h. Then, 100 µl of 10% SDS in 0.01 M HCl was added to each well. After further incubation at room temperature with shaking overnight, the plates were read at OD570 using Victor™ X3 Multilabel Reader (PerkinElmer, USA). Cell culture wells without peptides were used as the experiment control and medium only served as a blank control.

**Transmission electron microscopy assay**. To determine the effect of 8P9R on viral particles, SARS-CoV-2 was pretreated by 50 µg ml$^{-1}$ of 8P9R, P9R or P9RS for 1 h. The virus was fixed by formalin for overnight and then applied to continuous carbon grids. The grids were transferred into 4% uranyl acetate and incubated for 1 min. After removing the solution, the grids were air-dried at room

temperature. For each sample, two to three independent experiments were done for taking images by transmission electron microscopy (FEI Tecnal G2-20 TEM).

**Virus fluorescence assay**. To identify the effect of 8P9R on virus, H1N1 virus or SARS-CoV-2 was pre-labelled by green Dio dye (Invitrogen, Cat#3898) according to the manufacture introduction. Dio-labeled virus was treated by 8P9R, P9RS, or P9R (25 µg ml⁻¹) for 45 min. MDCK or Calu-3 cells were infected by the pre-treated virus for 1 h. Virus and cells were fixed by 4% formalin. Cell membrane was stained by membrane dye Alexa 594 (red, Invitrogen, W11262) and cell nucleus were stained by DAPI (blue). Virus entry or without entry on cell membrane was determined by confocal microscope (Carl Zeiss LSM 700, Germany).

**Endosomal acidification assay**. Endosomal acidification was detected with a pH-sensitive dye (pHrodo Red dextran, Invitrogen, Cat#P10361) according to the manufacturer's instructions with slight modification[43]. First, MDCK cells were treated with BSA (25.0 µg ml⁻¹), 8P9R (25.0 µg ml⁻¹), bafilomycin A1 (50.0 nM) at 4 °C for 15 min. Second, MDCK cells were added with 100 µg ml⁻¹ of pH-sensitive dye and DAPI and then incubated at 4 °C for 15 min. Before taking images, cells were further incubated at 37 °C for 15 min and then cells were washed twice with PBS. Finally, PBS was added to cells and images were taken immediately with confocal microscope (Carl Zeiss LSM 700, Germany).

**Spike-ACE2 mediated cell fusion assay**. According to previous study[44], the pSpike of SARS-CoV-2, pACE2-human, or pGFP were transfected to 293T cells for protein expression. After 24 h, to trigger the spike-ACE2 mediated cell fusion, 293T-Spike-GFP cells were co-cultured with 293T-ACE2 cells with the supplement of drugs. The 293T-GFP cells were co-cultured with 293T-ACE2 cells as the negative control. For Huh-7 cell fusion assay, Huh-7 cells were co-cultured with 293T-spike-GFP with the supplement of drugs. Huh-7 cells were co-cultured with 293T-GFP cells as the negative control. After 8 h of co-culture, five fields were randomly selected in each well to take the cell fusion pictures by fluorescence microscopes.

**Western blot for TMPRSS2 digestion assay**. Calu-3 cells were pre-treated by camostat (Cam, 50 µg ml⁻¹), 8P9R (50 µg ml⁻¹) or PBS for 1 h and then S protein (1 µg) of SARS-CoV-2 was added to treated cells for 1 h to let TMPRSS2 to cleave S protein. S protein without cells and drugs was served as the no digestion control. Samples were lysed in 1× Passive Lysis Buffer (Promega, E194A) and applied to detect spike protein using anti-S2 (Sino biological, 40590-T62, 1:5000 dilution) and anti-rabbit HRP as secondary antibody (Thermo Scientific, 31460, 1:4000 dilution).

**Antiviral assay in animals**. BALB/c female mice (10–12 month old)[45] and female hamsters (6–8 week old)[46] were kept in biosafety level 2/3 laboratory (housing temperature between 22 and 25 °C with dark/light cycle) and given access to standard pellet feed and water ad libitum. All experimental protocols followed the standard operating procedures of the approved biosafety level 2/3 animal facilities. Animal ethical regulations were approved by the Committee on the Use of Live Animals in Teaching and Research of the University of Hong Kong[47]. To evaluate the drug toxicity in vivo, mice were intranasally inoculated with 8P9R (0.5 mg kg⁻¹), camostat (0.3 mg kg⁻¹), and orally inoculated with arbidol (30 mg kg⁻¹) and chloroquine (40 mg kg⁻¹). Two more doses were given to mice in the following day. Lung tissues were harvested at day 2 for H&E staining. The body-weight changes of drug-treated mice were monitored for 18 days. To evaluate the antiviral activity, mice/hamsters were intranasally inoculated with SARS-CoV or SARS-CoV-2 to lungs. At 8 h post infection, PBS, 8P9R, arbidol, chloroquine, camostat, or combinational drugs were given to animals. Two more doses were given to mice/hamsters in the following day. Viral loads in mouse/hamster lungs were measured at day 2 post infection by plaque assay.

**Reporting summary**. Further information on research design is available in the Nature Research Reporting Summary linked to this article.

## Data availability

All data that support the conclusions of the study are included in this paper. Source data are provided with this paper.

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

## Acknowledgements

This study was partly supported by the donations of Michael Seak-Kan Tong, the Shaw Foundation Hong Kong, Richard Yu and Carol Yu, May Tam Mak Mei Yin, Hong Kong Sanatorium & Hospital, Hui Ming, Hui Hoy and Chow Sin Lan Charity Fund Limited, Chan Yin Chuen Memorial Charitable Foundation, Marina Man-Wai Lee, the Hong Kong Hainan Commercial Association South China Microbiology Research Fund, the Jessie & George Ho Charitable Foundation, Perfect Shape Medical Limited, Kai Chong Tong, and Tse Kam Ming Laurence; and funding from the National Program on Key Research Project of China (grant no. 2020YFA0707500 and 2020YFA0707504). The funding sources had no role in the study design, data collection, analysis, interpretation, or writing of the report.

## Author contributions

H.Z. and K.Y. designed this study. H.Z., H.L., X.Z., Z.P., A.L., J.C., W.C., J.I., and A.W.H. C. performed experiments. K.T., J.C., C.C., M.Y., and A.Z. provided materials. H.Z., K.T., K.Y. interpreted the findings. H.Z., S.J., and K.Y. revised the paper.

## Competing interests

The authors declare no competing interests.

## Additional information

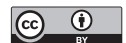

