## [Peer Review File · Nature Communications]

Reviewer comments, first round –

Reviewer #1 (Remarks to the Author):

Two distinct entry pathways for SARS-CoV-2 have been previously reported: a TMPRSS2-dependent plasma membrane pathway and a cathepsin-L-dependent endosome pathway. Zhao et al demonstrate that a cross-linking peptide 8P9R inhibits both entry pathways. By using VeroE6 cells, which have only the endosome pathway, the authors found that endosomal acidification inhibitors (8P9R or chloroquine) can significantly enhance the antiviral activity of arbidol to block the endosome pathway. In addition to endosomal acidification, 8P9R targets virus to reduce its infectivity probably by cross-linking to form viral clusters, thereby cooperating with arbidol to block the endosome pathway. By using Calu-3 cells, cells of lung epithelial origin that have only the plasma membrane pathway, they show that 8P9R can also block the plasma membrane pathway and that neither arbidol nor chloroquine affects the plasma membrane pathway. Consistently, in vivo SARS-CoV-2 infection of mice and hamsters was reduced by 8P9R and the drug combination including arbidol, chloroquine and camostat (TMPRSS2 inhibitor) but not by any single drug or any two-drug combination. Although the idea of blocking both entry pathways to reduce viral load in vivo is not entirely new, the cross-linking peptide 8P9R is interesting and may prove useful for treating COVID-19. Therefore, further experiments that could provide additional information about the molecular mechanisms of 8P9R action are needed for publication in this journal.

Major comments

- 1) The ability of 8P9R to inhibit SARS-CoV-2 infection in vitro is shown in Fig. 1 c-f using VeroE6 cells, which have only the endosome entry pathway. To understand how 8P9R inhibits the TMPRSS2-dependent entry, experiments shown in Fig.1 c-f should also be performed using Calu-3 cells, in which the TMPRSS2-dependent pathway dominates.
- 2) It would be interesting to test whether 8P9R inhibits TMPRSS2 activity or S protein priming.
- 3) Fig. 2a; Please quantitate cluster formation of the virus in terms of size and number.
- 4) Fig. 2b & Supplementary Fig. 4; Similar experiments should be performed using Calu-3 with SARS-CoV-2.
- 5) Based on the results of 1)-4) described above, together with the results in the original manuscript, the authors may then discuss in more detail the molecular mechanism of 8P9R-mediated inhibition of the TMPRSS2-dependent entry pathway.

Minor comments

- 1) Page 7, lines 139-140; (Supplementary Fig. 6) is redundant.
- 2) Page 8, lines 158-159; please explain why pH in the endosome affects fusion that occurs at the plasma membrane. Related question to page 11, lines 231-235; please explain why you think that your results regarding the effect of endosomal pH on plasma membrane fusion is consistent with the paper about the effects of endosomal pH on the membrane fusion of pseudoviruses with endosomes.
- 3) Page 9, line 180; 25 mg should read as 30 mg.
- 4) Fig. 3e; "+Spike+R8-25" should read as "+Spike+R8P9R-25"?

Reviewer #2 (Remarks to the Author):

This is a review of the manuscript entitled, "Cross-linking peptide and repurposed 1 drugs inhibit both entry pathways of SARS-CoV-2" by Zhao et al. In this interesting and ambitious pilot study, the authors combine several experimental treatments mostly in vitro, to suggest synergistic activity for possible use in vivo.

Major comments:

This is a study that has an interesting premise and initial trajectory. However there seems to be

hastily made towards combination therapy without fully validating the efficacy and safety of each test article in vitro or whether there are adverse effects when used in combination. Is a hemolysis of turkey RBCs the best approach for assessment of cytotoxicity? The in vivo studies seemed to be superficial and rushed and lack content as to why these were done as performed. Were tissues of mice and hamsters screened to evaluated for cytotoxicity in vivo? Were there lesion differences between the various therapies? The methods for the mouse experiments could made more clear. Why were 10 month old mice and 6 week old hamsters used in the study? Why were they all female? Seems like a random rather than a planned strategy. What about sex differences seen in mice (PMID: 28373583) and humans (PMID: 32846427)? If male mice were used would these results change?

Fig. 2A – This could easily be quantified to validate the images.

Fig 3A – it would make more sent to list the order of the intragraph legend to match the lines and concentrations, So blue, red and then orange.

Fig 3B – The asterisks suggest there is a statistical difference between the last two bars, and also a 3 fold difference? This does not seem possible by the numbers on the y axis, please double check or clarify.

Fig 3E – These images are not very compelling, and quantification should be performed to validate any interpretations.

Fig 4A, B – what are these graphs measuring (i.e infection of what?) – the figure legend needs to be clarified.

Fig 4 – the animal experiments are not well described here – clarify for the reader what tissues are you measuring.

Reviewer #3 (Remarks to the Author):

In this study, Zhao et al. showed data to demonstrate that branched peptide 8P9R, from their previous published P9 and P9R, could cross-link viral particles with very potent antiviral activity. They used two different methods to show the cross-linking activity of 8P9R for clustering viruses. 8P9R can cross-link viruses to stick viruses on cell membrane without entry and prevent viral entry through endocytic pathway by inhibiting endosomal acidification. This is novel and very interesting. They further demonstrated that 8P9R could efficiently inhibit SARS-CoV-2 replication in Vero E6 (SARS-CoV-2 entry through endocytic pathway) and Calu-3 cells (SARS-CoV-2 entry through TRMPRSS-2 mediated pathway), which indicated that 8P9R could block the two entry pathways of SARS-CoV-2. Also, 8P9R can significantly inhibit SARS-CoV-2 in hamsters and SARS-CoV in mice. The results indicated that 8P9R blocking the two entry pathways of SARS-CoV-2 could inhibit viral infection in vitro and in vivo. Moreover, the authors identified that endosomal acidification inhibitors (8P9R and chloroquine) could enhance the antiviral activity of arbidol and then they used three clinical inhibitors which can inhibit SARS-CoV-2 entry through endocytic pathway (arbidol and chloroquine) and TMPRSS2-mediated pathway (camostat) to confirm that blocking the two entry pathways of SARS-CoV-2 could significantly inhibit SARS-CoV and SARS-CoV-2 in animals. These clinical drug data provide important information for COVID-19 treatment, although the single use of the clinical drug did not show clinical benefits to patients in clinical trials. The overall experiment designs are well. Results and discussion are presented in the reasonable ways.

Major points:

1, In line 100, what is P9RS? It should be an important control from the author's published study. It is better to explain it more clearly here.

2, In Fig.2ab, authors showed that 8P9R could efficiently cross-link SARS-CoV-2 and H1N1 viruses. The data are solid to support the conclusion. However, what is the possible mechanism of 8P9R binding to different viruses?

3, In Fig. 4f, when authors used intranasal inoculation with camostat for treating SARS-CoV-2, which might be more efficiently than oral administration. Because this drug is normally used by oral administration in patients, it may provide more information if the authors can test the antiviral efficiency of camostat for SARS-CoV-2 in hamsters by oral administration.

4. The detailed information of SARS-CoV-2 gene in the virus strain used is needed. Is it different from the sequence of the original strain?

5. In each of the panels of the Main and Supplemental Figures, the authors need to indicate the number of independent experiments (biological replicates) and technical replicates within an experiment. Moreover, for data that is analyzed statistically, they need to indicate that it is derived from pooled data from the independent experiments.

6. The vast majority of the statistical analyses is across three or more groups and requires an ANOVA rather than student's t test. This should be corrected throughout the Main and Supplemental Figures.

Minor:

1, In line 80, 'comastat' should be camostat

2, Fig. 1c, the label in X axis, '1.8' should be 0.8

3, Fig. 4c, the label in Y axis, 'PUF' should be PFU

Point-to-point response to the reviewers

Reviewer #1 (Remarks to the Author):

Two distinct entry pathways for SARS-CoV-2 have been previously reported: a TMPRSS2-dependent plasma membrane pathway and a cathepsin-L-dependent endosome pathway. Zhao et al demonstrate that a cross-linking peptide 8P9R inhibits both entry pathways. By using VeroE6 cells, which have only the endosome pathway, the authors found that endosomal acidification inhibitors (8P9R or chloroquine) can significantly enhance the antiviral activity of arbidol to block the endosome pathway. In addition to endosomal acidification, 8P9R targets virus to reduce its infectivity probably by cross-linking to form viral clusters, thereby cooperating with arbidol to block the endosome pathway. By using Calu-3 cells, cells of lung epithelial origin that have only the plasma membrane pathway, they show that 8P9R can also block the plasma membrane pathway and that neither arbidol nor chloroquine affects the plasma membrane pathway. Consistently, in vivo SARS-CoV-2 infection of mice and hamsters was reduced by 8P9R and the drug combination including arbidol, chloroquine and camostat (TMPRSS2 inhibitor) but not by any single drug or any two-drug combination. Although the idea of blocking both entry pathways to reduce viral load in vivo is not entirely new, the cross-linking peptide 8P9R is interesting and may prove useful for treating COVID-19. Therefore, further experiments that could provide additional information about the molecular mechanisms of 8P9R action are needed for publication in this journal.

Major comments

1) The ability of 8P9R to inhibit SARS-CoV-2 infection in vitro is shown in Fig. 1 c-f using VeroE6 cells, which have only the endosome entry pathway. To understand how 8P9R inhibits the TMPRSS2-dependent entry, experiments shown in Fig.1 c-f should also be performed using Calu-3 cells, in which the TMPRSS2-dependent pathway dominates.

Response:

Thank you for the comments.

Figure 1c shows that the antiviral effect when 8P9R was premixed with SARS-CoV-2, and the plaque reduction assay was performed on VeroE6 cells. Since Calu-3 cells are not suitable for plaque assay, we could not do plaque reduction assay on Calu-3 cells.

Figure 1d shows the antiviral effect when 8P9R was mixed with SARS-CoV-2 at the time of infection in VeroE6 cells. We used Calu-3 cells to show the antiviral activity of 8P9R at the time of viral infection against SARS-CoV-2 in Figure 4e.

Figure 1E shows the antiviral effect when 8P9R was added to SARS-CoV-2 infected VeroE6 cells at 6 hours post-infection. We have now performed the similar experiment for Calu-3 cells and shown the results in the new Supplementary Figure 10.

Fig. 1f is RBC hemolysis assay for 8P9R. From these data in Fig. 4e and supplementary Fig. 10, we showed that 8P9R could significantly inhibit SARS-CoV-2 replication in Calu-3 cells.

2) It would be interesting to test whether 8P9R inhibits TMPRSS2 activity or S protein priming.

Response:

We would like to thank the reviewer for the suggestion. We did further experiments to test if 8P9R could inhibit TMPRSS2 activity to proteolytically activate the spike protein (Supplementary Fig. 11 in line 193). When compared with positive control camostat, 8P9R did not inhibit TMPRSS2 in cleaving S protein. This indicated that the inhibition of SARS-CoV-2 growth in Calu-3 cells by 8P9R was not attributed to the inhibition on TMPRSS2 activity. Instead, the inhibition of 8P9R on SARS-CoV-2 in Calu-3 cells was mainly related to the physical clustering of viruses to big clumps which blocked the viral entry step (shown in new supplementary Fig. 12 in line 197).

3) Fig. 2a; Please quantitate cluster formation of the virus in terms of size and number.

Response:

We would like to thank the reviewer for the suggestion. We have now quantitated the number of non-clustered viral particles and clustered viral particles (see Figure 2 legend from line 565 to 570). 'For quantification, 55 independent viral particles of P9RS-treated virus, 50 independent viral particles of P9R-treated virus, and 13 viral particles (including independent and clustered particles) of 8P9R-treated virus could be accounted in 5 representative microscope fields. The big clustering viral particles in 8P9R-treated samples could be more than 500 nm, which was bigger than the size (~100 nm) of the usual SARS-CoV-2 virion'.

4) Fig. 2b & Supplementary Fig. 4; Similar experiments should be performed using Calu-3 with SARS-CoV-2.

Response:

Thank you for this comment. We have now performed new experiments on Calu-3 cells (Supplementary Fig. 12 in line 197). The data showed that 8P9R could aggregate SARS-CoV-2 on Calu-3 membrane without entry. The size of the aggregated SARS-CoV-2 clumps were >500 nm, which was bigger than the usual SARS-CoV-2 (~100 nm) virion. This is consistent with the antiviral activity of 8P9R against SARS-CoV-2 in Calu-3 cells.

5) Based on the results of 1)-4) described above, together with the results in the original manuscript, the authors may then discuss in more detail the molecular mechanism of 8P9R-mediated inhibition of the TMPRSS2-dependent entry pathway.

Response:

We would like to thank the reviewer for the suggestions. In the results section, we added the findings in line 192-197: ‘We further demonstrated that camostat but not 8P9R could inhibit TMPRSS2 activity to cleave S protein (Supplementary Fig. 11), which indicated that the inhibition of SARS-CoV-2 replication in Calu3 cells was not related to the inhibition of TMPRSS2 activity. Instead, the inhibition of SARS-CoV-2 replication in Calu-3 cells by 8P9R was attributed to the physical clustering of viral particles which resulted in blocking of viral entry (Supplementary Fig. 12)’. In the discussion section (from line 245-248), we have now amended the following sentences: ‘In contrast, 8P9R could significantly inhibit coronaviruses *in vivo*. 8P9R not only blocked the endocytic pathway by preventing endosomal acidification, but also caused the formation of big clumps of aggregated viral particles which could no longer enter cells by direct membrane fusion in Calu-3 cells (Supplementary Fig. 12)’.

Minor comments

1) Page 7, lines 139-140; (Supplementary Fig. 6) is redundant.

Response:

Thank you for this comment. We removed the sentence and Fig. S6.

2) Page 8, lines 158-159; please explain why pH in the endosome affects fusion that occurs at the plasma membrane. Related question to page 11, lines 231-235; please explain why you think that your results regarding the effect of endosomal pH on plasma membrane fusion is consistent with the paper about the effects of endosomal pH on the membrane fusion of pseudoviruses with endosomes.

Response:

Thank you for this comment. According to the results of endosomal acidification inhibitors (bafilomycin A1, NH₄Cl and chloroquine) which inhibited spike-ACE2 mediated fusion, we suspected that this fusion process should happen at the endocytic pathway, and not only fusion at the plasma membrane without endocytosis. Pseudovirus entry was mediated by spike-ACE2 binding and the endocytic pathway for luciferase expression, which could be inhibited by NH₄Cl.

Thus, the fusion conditions between our spike-ACE2 and the pseudovirus entry should be the same, which were both mediated by the spike-ACE2 binding.

3) Page 9, line 180; 25 mg should read as 30 mg.

Response:

Thanks for this correction. We corrected it to 30 mg.

4) Fig. 3e; “+Spike+R8-25” should read as “+Spike+R8P9R-25”?

Response:

Thank you for this correction. We corrected it to +Spike+8P9R-25.

Reviewer #2 (Remarks to the Author):

This is a review of the manuscript entitled, "Cross-linking peptide and repurposed 1 drugs inhibit both entry pathways of SARS-CoV-2" by Zhao et al. In this interesting and ambitious pilot study, the authors combine several experimental treatments mostly in vitro, to suggest synergistic activity for possible use in vivo.

Major comments:

This is a study that has an interesting premise and initial trajectory. However there seems to be hastily made towards combination therapy without fully validating the efficacy and safety of each test article in vitro or whether there are adverse effects when used in combination. Is a hemolysis of turkey RBCs the best approach for assessment of cytotoxicity? The in vivo studies seemed to be superficial and rushed and lack content as to why these were done as performed. Were tissues of mice and hamsters screened to evaluated for cytotoxicity in vivo? Were there lesion differences between the various therapies? The methods for the mouse experiments could made more clear. Why were 10 month old mice and 6 week old hamsters used in the study? Why were they all female? Seems like a random rather than a planned strategy. What about sex differences seen in mice (PMID: 28373583) and humans (PMID: 32846427)? If male mice were used would these results change?

Response:

We would like to thank the reviewer for the comment. Regarding toxicity, in our original submission, we tested the toxicity of 8P9R *in vitro* using the MTT assay (Supplementary Fig. 2) and hemolysis assay (Fig. 1f). Hemolysis assay is acceptable for toxicity testing *in vitro*^{1,2}. We have now added new experiments to determine the toxicity/safety of 8P9R and the triple combination of arbidol-camostat-chloroquine in mice (new Supplementary Figure 9a-9b). No obvious toxicity was observed in the H&E staining of lung tissues harvested from 8P9R or drug combination treated mice (Supplementary Figure 9a). Body weight changes were monitored for 18 days after drug administration (Supplementary Fig. 9b). Body weight was slightly decreased in PBS, 8P9R and three-drug combination groups at day 1. All the body weights started to recover at day 2 after stopping drug administration and then the body weight changes in all groups were similar to that of the naïve group, which indicated that these drugs did not show obvious toxicity/safety problem at the administrated dosage *in vivo*. The toxicity and efficacy of old clinical drug arbidol, chloroquine and camostat had been tested by previous studies^{3,4}. Accordingly, we selected the no-toxicity concentration *in vitro* for this study. For animal experiments, we referred to previous studies⁵⁻⁸ in our selection of treatment doses of arbidol, chloroquine, and camostat for mice and hamsters in this study. The single drug did not inhibit SARS-CoV and SARS-CoV-2 replication *in vivo*. The triple combination of arbidol, chloroquine, and camostat could significantly inhibit viral replication *in vivo*.

We selected old mice (>9 months) for SAR-CoV in this study, which was because the previous study showed that SARS-CoV replicated better in old mice than that in young mice⁹. For SARS-CoV-2, we used hamsters of 6-8 weeks old because our group had previously tested SARS-CoV-2 in hamsters of this age¹⁰. The replication of SARS-CoV could reach the peak titer at day 2 post infection and mice started to recover after day 5 post infection⁹. Thus, we selected day 2 post infection to collect the tissues to evaluate the antiviral activity of drugs for SARS-CoV in mice. SARS-CoV-2 replication reached the peak titer at day 2 post infection and hamsters started to recover after day 5-7 post infection¹⁰. Thus, we selected the same time point as SARS-CoV at day 2 post infection to evaluate the antiviral activity of drugs inhibiting viral replication in lungs. We did not collect the lung tissues for histological testing, which was because our hamster model¹⁰ showed slight inflammation in lungs at day 2 post-infection even with high viral load inoculation (1×10^5 PFU).

The reviewer is correct that there may be differences in response to drugs with different sex. This was the reason why we limited our investigations to female to minimize confounding effects due to sex differences. It would be interesting to investigate sex and age effects on drug antiviral activities against SARS-CoV-2 in different animal models.

1. Holthausen, D.J. et al. An Amphibian Host Defense Peptide Is Virucidal for Human H1 Hemagglutinin-Bearing Influenza Viruses. *Immunity* 46, 587-595 (2017).
2. Zhu, S., Gao, B., Harvey, P.J. & Craik, D.J. Dermatophytic defensin with antiinfective potential. *Proc Natl Acad Sci U S A* 109, 8495-8500 (2012).
3. Wang, M. et al. Remdesivir and chloroquine effectively inhibit the recently emerged novel coronavirus (2019-nCoV) in vitro. *Cell Res* (2020).
4. Hoffmann, M. et al. SARS-CoV-2 Cell Entry Depends on ACE2 and TMPRSS2 and Is Blocked by a Clinically Proven Protease Inhibitor. *Cell* 181, 271-280 e278 (2020).
5. Shi, L. et al. Antiviral activity of arbidol against influenza A virus, respiratory syncytial virus, rhinovirus, coxsackie virus and adenovirus in vitro and in vivo. *Arch Virol* 152, 1447-1455 (2007).
6. Zhou, Y. et al. Protease inhibitors targeting coronavirus and filovirus entry. *Antiviral Res* 116, 76-84 (2015).
7. Freiberg, A.N., Worthy, M.N., Lee, B. & Holbrook, M.R. Combined chloroquine and ribavirin treatment does not prevent death in a hamster model of Nipah and Hendra virus infection. *J Gen Virol* 91, 765-772 (2010).
8. Kaptein, S.J.F. et al. Favipiravir at high doses has potent antiviral activity in SARS-CoV-2-infected hamsters, whereas hydroxychloroquine lacks activity. *Proc Natl Acad Sci U S A* 117, 26955-26965 (2020). (<https://doi.org/10.1101/2020.06.19.159053>)
9. Roberts, A. et al. Aged BALB/c mice as a model for increased severity of severe acute respiratory syndrome in elderly humans. *Journal of virology* 79, 5833-5838 (2005).
10. Chan, J.F. et al. Simulation of the clinical and pathological manifestations of Coronavirus Disease 2019 (COVID-19) in golden Syrian hamster model: implications for disease pathogenesis and transmissibility. *Clin Infect Dis* (2020).

Fig. 2A – This could easily be quantified to validate the images.

Response:

Thank you for this comment. We quantified the clustered virus particles and single viral particles in Fig. 2 legend (line 565-570): ‘For quantification, 55 independent viral particles of P9RS-treated virus, 50 independent viral particles of P9R-treated virus, and 13 viral particles (including independent and clustered particles) of 8P9R-treated virus could be accounted in 5 representative microscope fields. The big clustering viral particles in 8P9R-treated samples could be more than 500 nm, which was bigger than the size (~100 nm) of the usual SARS-CoV-2 virion’.

Fig 3A – it would make more sense to list the order of the intragraph legend to match the lines and concentrations, So blue, red and then orange.

Response:

Thank you for this comment. We revised it to blue-red-orange in Fig. 3a accordingly.

Fig 3B – The asterisks suggest there is a statistical difference between the last two bars, and also a 3 fold difference? This does not seem possible by the numbers on the y axis, please double check or clarify.

Response:

Thank you for this comment. We double checked Fig. 3b and noted that $8P9R=3.1=125.1$ and $Ar+8P9R=38.2$ The fold was $125.1/38.2=3.3$.

Fig 3E – These images are not very compelling, and quantification should be performed to validate any interpretations.

Response:

Thank you for this comment. We revised the figures and quantified the fused cells as described in the figure 3E legend (in line 613-614): 8P9R ($25 \mu\text{g ml}^{-1}$) and arbidol ($25 \mu\text{g ml}^{-1}$) did not block cell fusion, of which the fused cells were 2-10-fold bigger than the non-fused cells.

Fig 4A, B – what are these graphs measuring (i.e infection of what?) – the figure legend needs to be clarified.

Response:

Thank you for this comment. We revised it in the Fig. 4 legend (line 642-644): ‘Viral RNA copies were measured at 24 h post infection in cell supernatants. The relative RNA copy was compared to mock treated virus’.

Fig 4 – the animal experiments are not well described here – clarify for the reader what tissues are you measuring.

Response:

Thank you for this comment. We revised the methods for animal experiment in the figure 4 legend (line 649-650): ‘Two more doses were given to mice in the following day. Viral loads in lung tissues were measured by plaque assay at day 2 post infection.’ and in line 654-659: ‘Hamsters were intranasally inoculated with SARS-CoV-2 (5×10^3 PFU). Mock (n=9), 8P9R (intranasal 0.5 mg kg^{-1} , n=4), Ar+Chl+Cam (n=6), Chl+Cam (n=6), Ar+Cam (n=3), Cam (intranasal 0.3 mg kg^{-1} , n=5), Ar (oral 25 mg kg^{-1} , n=3), and Chl (oral 40 mg kg^{-1} , n=4) were

given to hamsters at 8 h post infection. Two more doses were given to hamsters in the following day. Viral loads in lung tissues were measured by plaque assay at day 2 post infection’.

Reviewer #3 (Remarks to the Author):

In this study, Zhao et al. showed data to demonstrate that branched peptide 8P9R, from their previous published P9 and P9R, could cross-link viral particles with very potent antiviral activity. They used two different methods to show the cross-linking activity of 8P9R for clustering viruses. 8P9R can cross-link viruses to stick viruses on cell membrane without entry and prevent viral entry through endocytic pathway by inhibiting endosomal acidification. This is novel and very interesting. They further demonstrated that 8P9R could efficiently inhibit SARS-CoV-2 replication in Vero E6 (SARS-CoV-2 entry through endocytic pathway) and Calu-3 cells (SARS-CoV-2 entry through TRMPRSS-2 mediated pathway), which indicated that 8P9R could block the two entry pathways of SARS-CoV-2. Also, 8P9R can significantly inhibit SARS-CoV-2 in hamsters and SARS-CoV in mice. The results indicated that 8P9R blocking the two entry pathways of SARS-CoV-2 could inhibit viral infection in vitro and in vivo. Moreover, the authors identified that endosomal acidification inhibitors (8P9R and chloroquine) could enhance the antiviral activity of arbidol and then they used three clinical inhibitors which can inhibit SARS-CoV-2 entry through endocytic pathway (arbidol and chloroquine) and TMPRSS2-mediated pathway (camostat) to confirm that blocking the two entry pathways of SARS-CoV-2 could significantly inhibit SARS-CoV and SARS-CoV-2 in animals. These clinical drug data provide important information for COVID-19 treatment, although the single use of the clinical drug did not show clinical benefits to patients in clinical trials. The overall experiment designs are well. Results and discussion are presented in the reasonable ways.

Major points:

1, In line 100, what is P9RS? It should be an important control from the author’s published study. It is better to explain it more clearly here.

Response:

Thank you for this comment. We now revised the description in line 100: ‘P9RS which was a basic peptide with no antiviral activity or ability to bind virus’.

2, In Fig.2ab, authors showed that 8P9R could efficiently cross-link SARS-CoV-2 and H1N1

viruses. The data are solid to support the conclusion. However, what is the possible mechanism of 8P9R binding to different viruses?

Response:

Thank you for this comment. According to our previous study in Nat Communications paper¹¹, P9R did not show a fixed NMR structure in the water solution. We hypothesized that the flexible structure of P9R could allow P9R to change its structure in order to fit the target protein for better binding. We are interested in this broadly binding ability of P9R, which will need further co-binding structure analysis.

3, In Fig. 4f, when authors used intranasal inoculation with camostat for treating SARS-CoV-2, which might be more efficiently than oral administration. Because this drug is normally used by oral administration in patients, it may provide more information if the authors can test the antiviral efficiency of camostat for SARS-CoV-2 in hamsters by oral administration.

Response:

Thank you for this comment. We performed new experiments to evaluate the antiviral activity of camostat by oral administration (Supplementary Fig. 14 in line 209). Camostat did not show antiviral activity but the triple combination (Cam+Ar+Chl) could inhibit SARS-CoV-2 replication in hamster lungs when drugs were orally administrated.

4. The detailed information of SARS-CoV-2 gene in the virus strain used is needed. Is it different from the sequence of the original strain?

Response:

Thank you for this comment. We sequenced the SARS-CoV-2 used in this study. We could only detect a short deletion in S protein and mentioned it in the material and method in line 276. This was also reported by other group members with the same deletion in SARS-CoV-2.

5. In each of the panels of the Main and Supplemental Figures, the authors need to indicate the number of independent experiments (biological replicates) and technical replicates within an experiment. Moreover, for data that is analyzed statistically, they need to indicate that it is derived from pooled data from the independent experiments.

Response:

Thank you for this comment. We now indicated the numbers of independent biological samples in all figure legends. No pooled data were used in this study.

6. The vast majority of the statistical analyses is across three or more groups and requires an ANOVA rather than student's t test. This should be corrected throughout the Main and Supplemental Figures.

Response:

Thank you for this comment. When comparing more than two groups, an ANOVA test should be used. In this study, all *P* values were generated from comparing two groups. We did not compare three groups to get *P* values in this study. We noted that the *P* values were confusing in Fig. 4d. We have revised the figure to show *P* values clearly.

Minor:

1, In line 80, 'comastat' should be camostat

Response:

Thank you for this correction. We corrected it to camostat.

2, Fig. 1c, the label in X axis, '1.8' should be 0.8

Response:

Thank you for this correction. We corrected it to 0.8.

3, Fig. 4c, the label in Y axis, 'PUF' should be PFU

Response:

Thank you for this correction. We corrected it to PFU.

1. Holthausen, D.J. et al. An Amphibian Host Defense Peptide Is Virucidal for Human H1 Hemagglutinin-Bearing Influenza Viruses. *Immunity* **46**, 587-595 (2017).
2. Zhu, S., Gao, B., Harvey, P.J. & Craik, D.J. Dermatophytic defensin with anti-infective potential. *Proc Natl Acad Sci U S A* **109**, 8495-8500 (2012).
3. Wang, M. et al. Remdesivir and chloroquine effectively inhibit the recently emerged novel coronavirus (2019-nCoV) in vitro. *Cell Res* (2020).
4. Hoffmann, M. et al. SARS-CoV-2 Cell Entry Depends on ACE2 and TMPRSS2 and Is Blocked by a Clinically Proven Protease Inhibitor. *Cell* **181**, 271-280 e278 (2020).

5. Shi, L. et al. Antiviral activity of arbidol against influenza A virus, respiratory syncytial virus, rhinovirus, coxsackie virus and adenovirus in vitro and in vivo. *Arch Virol* **152**, 1447-1455 (2007).
6. Zhou, Y. et al. Protease inhibitors targeting coronavirus and filovirus entry. *Antiviral Res* **116**, 76-84 (2015).
7. Freiberg, A.N., Worthy, M.N., Lee, B. & Holbrook, M.R. Combined chloroquine and ribavirin treatment does not prevent death in a hamster model of Nipah and Hendra virus infection. *J Gen Virol* **91**, 765-772 (2010).
8. Kaptein, S.J.F. et al. Favipiravir at high doses has potent antiviral activity in SARS-CoV-2-infected hamsters, whereas hydroxychloroquine lacks activity. *Proc Natl Acad Sci U S A* **117**, 26955-26965 (2020).
9. Roberts, A. et al. Aged BALB/c mice as a model for increased severity of severe acute respiratory syndrome in elderly humans. *Journal of virology* **79**, 5833-5838 (2005).
10. Chan, J.F. et al. Simulation of the clinical and pathological manifestations of Coronavirus Disease 2019 (COVID-19) in golden Syrian hamster model: implications for disease pathogenesis and transmissibility. *Clin Infect Dis* (2020).
11. Zhao, H. et al. A broad-spectrum virus- and host-targeting peptide against respiratory viruses including influenza virus and SARS-CoV-2. *Nat Commun* **11**, 4252 (2020).

Reviewer comments, second round –

Reviewer #1 (Remarks to the Author):

I would like to thank the authors for their response to my comments. I am content with the adaptations, which have certainly improved the manuscript. The revised ms is suitable for publication.

Reviewer #2 (Remarks to the Author):

Thank you for clarifying and addressing my comments

Reviewer #3 (Remarks to the Author):

I do not have any new comments

Point-to-point response

REVIEWERS' COMMENTS

Reviewer #1 (Remarks to the Author):

I would like to thank the authors for their response to my comments. I am content with the adaptations, which have certainly improved the manuscript. The revised ms is suitable for publication.

Response:

Thanks for reviewer's comments.

Reviewer #2 (Remarks to the Author):

Thank you for clarifying and addressing my comments

Response:

Thanks for reviewer's comments.

Reviewer #3 (Remarks to the Author):

I do not have any new comments

Response:

Thanks for reviewer's comments.